# The Tasmanian Emergency Care Outcomes Registry (TECOR) Protocol

Viet Tran [1,2,3,*] , Giles Barrington [1] and Simone Page [1,2]

1    Royal Hobart Hospital, Tasmanian Health Service, Hobart 7000, Australia
2    Tasmanian School of Medicine, University of Tasmania, Hobart 7000, Australia
3    Menzies Institute for Medical Research, University of Tasmania, Hobart 7000, Australia
*    Correspondence: v.tran@utas.edu.au

**Abstract:** Emergency Departments (ED) play a vital role within the health system, representing the 'front door' to hospitals, the first point of hospital contact for patients who are undifferentiated and may be critically ill. They also serve as a safety net for the healthcare system. Together with ED overcrowding, this patient care environment is highly vulnerable to the provision of suboptimal care and breaches in patient safety. Government agencies in Australia currently collect data that are broad and administratively focused and are limited in capacity to identifying clinical quality. Clinical quality registries (CQR) help fill this gap but are often funded by not-for-profit organisations or research grants. There is no emergency care CQR in Tasmania, Australia. We propose the establishment of the Tasmanian Emergency Care Outcomes Registry (TECOR) to monitor emergency care processes and outcomes. The primary objective of TECOR is to monitor the unexpected 30-day mortality of patients who are cared for in the ED as well as 30-day safety events where emergency care was the primary contributor. The TECOR is expected to provide ongoing data on other important processes of emergency care in Tasmania such as length of stay in EDs, 28-day representation to EDs and hospital length of stay. The registry was designed to national standards and will meet the needs of the clinical community and have a positive impact on the communities it serves.

**Keywords:** emergency medicine; clinical registry; quality; safety; health service; outcomes; database

## 1. Introduction

### 1.1. The Nature and Safety of Emergency Medicine and Emergency Departments

Emergency Departments (ED) play a vital role within the health system. The Australasian College for Emergency Medicine (ACEM) is responsible for training emergency physicians and advancement of professional standards in emergency medicine in Australia and Aotearoa New Zealand [1]. ACEM defines an ED as 'a dedicated area in a hospital that is organised and administered to provide a high standard of emergency care to people in the community who perceive the need for, or are in need of, acute or urgent care including hospital admission' [2]. The ED often represents the 'front door' to hospitals, the first point of hospital contact for patients who are undifferentiated and may be critically ill. They also serve as a safety net for the healthcare system [3]. The ED is also a patient care environment that is highly vulnerable to the provision of suboptimal care and breaches in patient safety [4]. The reasons for this are multifactorial. Not only does the ED manage the highest proportion of undifferentiated patients in any hospital department, but these patients also range from the critically ill through to the worried well. This combination of undifferentiated and unwell patients places a significant burden on ensuring timely diagnosis and management to avoid significant adverse outcomes from care.

Staffing factors in the ED magnify this risk. Medical staffing in larger regional and metropolitan EDs in Australia is typically 4 tiered; (1) a small specialist workforce to manage patient flow and supervise junior doctors (2) doctors in training who rotate through a

variety of placements and sometimes a variety of EDs every 6–12 months (3) pre-vocational doctors (usually post graduate year 2–4) rotating throughout the hospital and occasioning the ED for 6–10 weeks at a time and requiring intermediate supervision and (4) intern medical officers (post graduate year 1) who rotate through ED as a mandatory 6 week term to complete general registration with the national medical board. The requirement to understand a broad range of diseases, the spectrum of symptoms that patients may present with, as well as the need to understand a broad range of processes for a short period of time is exhausting to attempt and difficult to achieve. A variety of solutions have been developed to try and counter these challenges, including guidelines, policies, and pathways. Despite these best intentions, EDs remain significantly unsafe [5].

Currently, the greatest risk to patient safety and the single greatest threat to the security of healthcare systems internationally is the exponential growth of overcrowding in EDs [6]. Although no universally agreed definition for ED overcrowding exists, it is generally accepted that it is 'a circumstance in which the demand for emergency services exceeds the capacity to deliver appropriate care within a reasonable time' [7–11]. ED overcrowding has consistently been shown to increase morbidity and mortality [12]. A variety of reasons are postulated for this and include reduced nurse-to-patient ratios leading to delays to therapy and reduced attention to the vulnerable (e.g., high falls risk patients), a reduction in clinical judgement due to suboptimal clinical settings (e.g., a brief physical examination in the waiting room) and increased times to be triaged, to be seen and to be transferred to the ward [13].

### 1.2. Routinely Collected Data in Australia

The Australian Institute of Health and Welfare (AIHW) is an independent statutory Australian Government agency that collects and manages data on health and welfare issues, including from state, territory and federal government agencies [14]. As its name suggests, the AIHW covers fields as diverse as housing assistance, homelessness, perinatal health, disability, cancer, hospital and hospital activity, alcohol and other drugs and mortality [14]. Included within this data collection are Emergency Department activity for all public hospitals in Australia as well as Emergency Department Care Activity for non-admitted patients [15,16] The level of data available for these datasets is broad and administratively focused (Table 1) [17]. Any clinical data are limited to diagnosis. In Australia, this diagnosis classification is a derivation of the International Classification of Disease system, with Australian modifications currently up to its 12th revision [18]. Further to this, in 2015, the Independent Health and Aged Care Pricing Authority (IHACPA) developed an abbreviated list for use in the ED known as the Emergency Department ICD-10-AM Principal Diagnosis Short List [19]. The quality of emergency care is therefore not captured in these datasets including a lack of reporting on morbidity or mortality related to emergency care.

### 1.3. Clinical Quality Registries

Whereas the premise of AIHW reporting is focused on safety and broad administrative indicators, clinical quality registries (CQRs) are focused on the quality of health care within specific clinical domains by systematically analysing health-related data for an eligible population [17,20]. The goal of CQRs is to improve health outcomes through benchmarking and reporting on clinical care key performance indicators with the intention of informing clinician behaviour and decision-making, reducing variations in practice and improving quality of care, health care outcomes and patient satisfaction [17,20–22].

Since government agencies have not funded CQRs, their establishment has largely occurred through not-for-profit organisations such as the Stroke Foundation or through grant funding [23–26].

The Australian Commission for Safety and Quality in Healthcare (ACSQH) is the peak body for recognising national CQRs [27]. Given the source of funding for most CQRs, it follows that most are related to disease outcomes with only a handful related to a medical specialty. The longest running specialty-based CQR in Australia is the Australian

and New Zealand Intensive Care Society Centre for Outcomes and Resources Evaluation (ANZICS CORE) [28]. Established in 1993, ANZICS CORE covers 99% of all Intensive Care Units in Australia and New Zealand, has published over 200 peer-reviewed papers and has delivered significant quality improvements within the specialty of intensive care including the development and implementation of the Australian and New Zealand Risk of Death model [28]. ANZICS CORE also continues to inform health policy and daily clinical practice while also providing infrastructure for measuring the translation of evidence into practice [29]. The Australian and New Zealand College of Anaesthetists received a grant in 2022 to establish a peri-operative CQR as the next emerging specialty-based CQR [30]. CQRs exist in Tasmania but only as part of nationally funded schemes and are often disease- or condition-based such as the stroke registry, fractured neck of femur registry and Australian Trauma Registry (Table 2) [23–26].

**Table 1.** Data fields for reporting in TECOR.

| Minimum Dataset | Additional Data |
| --- | --- |
| Age at presentation [1,2,3] | |
| Sex [1,2,3] | Episode of admitted patient care—length of stay in intensive care |
| Australian Postcode [1,2,3] | unit, total hours [1,2] |
| Geographic remoteness, remoteness classification [1,2] | Episode of admitted patient care—separation date [1,2] |
| Country of birth [1,2,3] | Episode of care—source of funding, patient funding source code [1,2] |
| Indigenous status [1,2,3] | Episode of admitted patient care—separation mode [1,2] |
| Hospital [1,2,3] | Episode of admitted patient care—number of days of |
| Mode of arrival [1,2,3] | hospital-in-the-home care [1,2] |
| Date of presentation [1,2,3] | Episode of care—principal diagnosis, code [1,2,5] |
| Time of presentation [1,2,3] | Episode of care—secondary diagnosis and beyond, code [1,2,5] |
| Date of triage [1,2,3] | Episode of admitted patient care—admission urgency status [1,2] |
| Time of triage [1,2,3] | Referred by [1,3] |
| ED end date [1,2,3] | Referred to, position, specialty [1] |
| ED end time [1,2,3] | Referred to time, date [1] |
| Service episode length (total min) [1,2,3] | Consultation, position, specialty [1] |
| Type of visit [1,2,3] | Consultation time, date [1] |
| Triage category [1,2,3] | Bed request time, date [1] |
| Date seen by medical officer [1,2,3] | Bed available time, date |
| Time seen by medical officer [1,2,3] | Departure Status [1,3] |
| Emergency Department wait time [1,2] | Clinical observations [4] |
| Disposition [1,2,3] | Pathology testing and blood products requests time, date and type [4] |
| ED stay—urgency-related group major diagnostic block [1,2] | Pathology testing results and blood products given time and date [4] |
| ED stay—principal diagnosis [1,2,3] | Imaging testing requests time, date and type [4] |
| ED stay—diagnosis classification type [1,2,3] | Imaging testing results time and date [4] |
| ED stay—physical departure date [1,2,3] | Discharge method [1,3] |
| ED stay—physical departure time [1,2,3] | Procedures performed [4] |
| Patient—compensable status [1,2,3] | Medications prescribed including route and dose [4] |
| ED stay—additional diagnosis, code [1,2,3] | Medical progress notes [1,4] |
| Episode of admitted patient care—admission date [1,2] | Nursing progress notes [1] |
| Episode of admitted patient care—admission time [1,2] | Goals of Care prior to arrival [1,4] |
| Safety event(s) | Goals of Care at admission [1,4] |
| URN [1,3] | Goals of Care at discharge [1,4] |
| Emergency Attendance ID [1,3] | Hospital Capacity [2] |
| ED triage/presenting complaint [1,3] | Emergency Department Capacity [1] |
| Admission time, date [1,2] | |

[1] Tasmanian Department of Health data available in auditable format; [2] AIHW mandatory fields; [3] Victorian Emergency Minimum Dataset; [4] Alfred Registry for Emergency Care Protocol; [5] Tasmanian Health Service reported ICD-10-AM 12th edition at time of protocol publication.

**Table 2.** Current observational trials in Tasmania that duplicate data collection.

| Observational Trial | Trial Description |
|---|---|
| PROCSED [1] | A quality improvement database on the current clinical practice of procedural sedation in the ED in order to improve the quality of care |
| EDNA [2] | The Emerging Drugs Network of Australia (EDNA) brings together emergency physicians, toxicologists and forensic laboratories to establish a standardised ED toxicosurveillance system in Australia [31] |
| ANZEDAR [2] | A binational airway registry prospectively capturing intubations and factors associated with first attempt success [32] |
| MET Call Database [1] | A quality improvement database on the MET calls in the ED to identify challenges to improved patient care |
| AuSCR [2] | The Australian Stroke Clinical Registry aims to provide national, prospective, systematic data on processes and outcomes for stroke [33] |
| ANZHFR [2] | A clinical quality registry that collects data about older people admitted to hospital with a broken hip in Australia and New Zealand [34] |
| ANZTR [2] | Focuses on monitoring trauma care, from the time of incident to discharge from definitive care, in order to reflect and act upon emerging trends and demands on the trauma system across Australia and New Zealand [24] |

EDNA Emerging Drugs Network of Australia; ANZEDAR Australia and New Zealand Emergency Department Airway Registry; MET Medical Emergency Team; AuSCR Australian Stroke Clinical Registry; ANZHFR Australian and New Zealand Hip Fracture Registry; ANZTR Australia and New Zealand Trauma Registry. [1] Emergency Department Quality Improvement Project. [2] National Registry.

### 1.4. Emergency Care Clinical Quality Registries

Internationally, the American College for Emergency Physicians has developed the Clinical Emergency Data Registry [35,36]. Similarly, Germany, where Emergency Medicine is yet to be recognised as a speciality, established the AKTIN (Action Alliance for Information and Communication Technology) emergency room register in 2013 [37]. The World Health Organisation also has an opt-in clinical registry for systematically collecting, aggregating and analysing case-based emergency care encounters in a simplified manner to appeal to low- and middle-income countries [38].

Despite the ACSQH recording 112 CQRs in their database, an emergency care registry is not among this list to inform the quality of clinical care provided [27]. Outside of the ACSQH CQR database, three local emergency care registries are reported in the peer review literature and include the Alfred Registry for Emergency Care (Alf-REC), the Rural Acute Hospital Data Register (RAHDaR) and Southwestern Sydney's Comprehensive Dataset for Research, Innovation and Collaboration (CEDRIC) [39–41]. Given the paucity of data for emergency care in Australia, the quality of care in Emergency Departments, including in Tasmania is therefore mostly unknown.

### 1.5. Australian and Tasmanian Emergency Care

In 2023, Australia recorded a population of over 26.8 million [42]. The Australian healthcare system adopts the universal health coverage model set out by the World Health Organization [43]. Over 8.80 million patients requiring care in the emergency department will attend a public ED each year at no cost [15].

Tasmania is an island state of Australia with a population of 572,000 people across 68,401 km$^2$ (26,410 mi$^2$) of land [42]. Tasmania has four public hospitals, all of which have a mixed (adult and paediatric) ED; Royal Hobart Hospital, Launceston General Hospital, Mersey Community Hospital and Northwest Regional Hospital (Figure 1a). Together they saw 178,292 presentations in 2021 [15]. Tasmanian EDs also have the unfortunate reputation for suffering some of the worst ED overcrowding in the nation, a catalyst for further harm that is unmeasured [44].

The Accessibility/Remoteness Index of Australia (ARIA), and the subsequent ARIA+, is a national model that quantifies spatial access to guide investment across a range of policies [45]. ARIA classifies access into the five categories of the Australian Statistical Geography Standard-Remoteness Area (ASGS-RA): RA1-Major cities of Australia; RA2-

Inner Regional Australia; RA3-Outer Regional Australia; RA4-Remote Australia; and RA5-Very Remote Australia [45]. By accounting for road distance and the population of cities, towns and communities, the ASGS-RA is further classified into the seven categories of the Modified Monash Model (MM) [46]. The Australian Government Department of Health has adopted the MM model to inform national policy [46]. Tasmania is the only state in Australia that does not have a major city (MM 1). It is entirely classified as regional or below, ranging from regional (MM 2) such as Hobart through to very remote communities (MM 7) such as Flinders' Island (Figure 1b). Given the remoteness, Tasmania does have district hospitals which provide a service for patients to be assessed and stabilised prior to retrieval to one of the four EDs. Although they do provide emergency care, district hospitals do not fit within the definition of an ED and are therefore not included in TECOR.

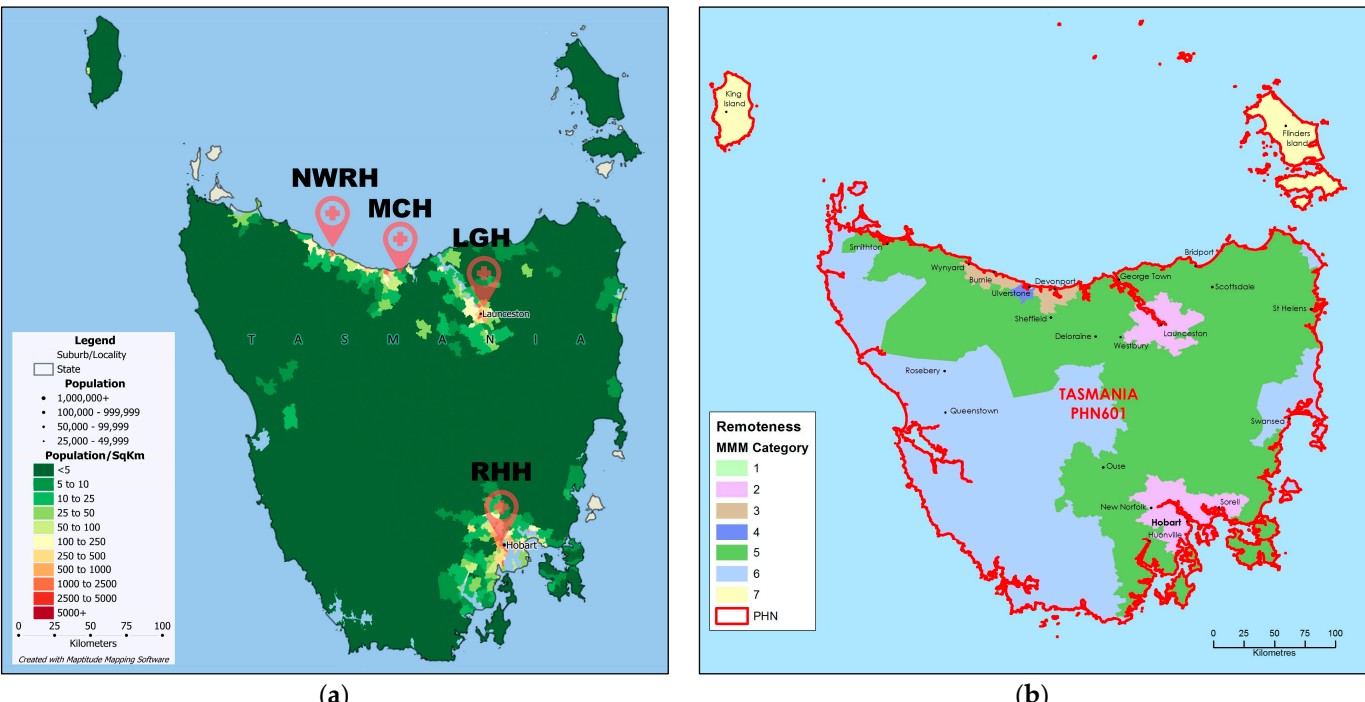

**Figure 1.** The Australian and Tasmanian Emergency Care Context: (**a**) Emergency Departments in Tasmania, Australia; NWRH North West Regional Hospital; MCH Mersey Community Hospital; LGH Launceston General Hospital; RHH Royal Hobart Hospital [47] (**b**) Modified Monash Model (MM) remoteness area classification for Tasmania, Australia. 1 Metropolitan areas; 2 Regional centres; 3 Large rural towns; 4 Medium rural towns; 5 Small rural towns; 6 Remote communities; Very remote communities [48].

In Tasmania, mortality data is linked to patient digital medical records. The Tasmanian Health Services utilises a safety reporting and learning system as a non-punitive, confidential and voluntary program to collect and analyse safety concerns by healthcare staff. These safety reports link to both patients and processes and are categorised based on the location of safety concerns.

### 1.6. Establishing the TECOR

Although a national ED CQR is needed, the establishment of such a CQR requires significant funding and inter-state and territory governance, data sharing and multi-institutional agreements. For now, a state-based ED CQR is more achievable and with time, may integrate with a funded national ED CQR. Further incentives for an emergency care-focused CQR in Tasmania include enhancing resilience and improving the capacity of EDs to care for all patients with acute illness and/or injury [49].

Tasmania, as an island state, is also well positioned to maintain a high rate of follow-up since most patients with an adverse outcome (death or recorded safety event) will likely have their ongoing care within one of the four hospitals included in TECOR.

## 2. Study Objectives

The primary objective of the TECOR is to monitor the unexpected 28-day mortality of patients who are cared for by the ED as well as 28-day safety events where emergency care was the primary contributor.

A suite of secondary objectives will consider outcomes for patients including length of stay in EDs, 28-day representation to EDs, hospital length of stay, in-hospital complications, and death; assess variation in practice and evaluate the impact of quality improvement initiatives over time; provide baseline data informing the design and conduct of quality improvement studies. It is noted that this list is not exhaustive and other areas of interest may arise as TECOR evolves (see Section 5.1 Future Nested Trials).

## 3. Materials and Methods

The TECOR will be established in accordance with the framework for Australian clinical quality registries and ensure compliance with the legislation and regulation relating to the clinical quality registries' final report, both published by the Australian Commission on Safety and Quality in Health Care [20,50].

The registry has been approved by the University of Tasmania Human Research Ethics Committee (HREA30260, 26 February 2024). The registry has also been publicly registered with the Australian and New Zealand Clinical Trial Registry (ACTRN12624000278538).

### 3.1. CQR Categorisation

TECOR is categorised as a research/record-keeping registry in accordance with the ACSQHC classification [50]. This category includes data repositories established as part of an approved Quality Assurance Activity or research study.

### 3.2. Selection of Hospitals

TECOR will include all four public EDs in the state (Figure 1a). It will not include district hospitals that provide emergency care as they are not by definition emergency departments [2].

### 3.3. Patient Eligibility

3.3.1. Inclusion Criteria

All patients presenting to a public Tasmanian ED. No individuals from a defined population (e.g., children, pregnant women, cultural identity) will be recruited into TECOR in a targeted manner by virtue of their being present in the general population from which the participants are being recruited (all ED presentations).

3.3.2. Exclusion Criteria

Patients who administratively registered as presenting to an ED but where emergency care was not actually provided by ED Staff, such as inter-hospital transfers that are required to enter the hospital via the ED for patient labels, will be excluded. Patients will also be excluded if they are not triaged as they will not have registered on the electronic system as being present. Patients who choose not to wait to be seen will be included if they are registered on the ED tracking system. Errors in duplication of patient presentation data will also be excluded. If a patient presents more than once a day, all presentations will act as separate entries in TECOR.

### 3.4. Data Collection

Routinely collected data will be retrospectively imported and prospectively collected monthly. Retrospective data will be from the 1st of January 2016 when the Australian

Institute of Health and Welfare minimum data sets began reporting and will inform trends and benchmark clinical quality. Data fields consist of routinely collected data by the Tasmanian Department of Health for AIHW reporting, as well as clinical data based on the Victorian Emergency Minimum Dataset and Alfred Registry for Emergency Care (Table 1) [16,51,52]. Data will be collected via semi-automated importation into the registry for routinely collected data and manually inputted where automation is not possible. Individual records within the registry will be assigned a separate registry number. Registry numbers and their corresponding patient unique record number (URN) will be stored separately from the registry data on the Department of Health, Tasmania servers with separate password-protected access by the research team only to allow re-identification when required. Clinicians not named on the ethics and governance application will not have access to re-identifiable data. Should clinicians or researchers require access to re-identifiable data, amendments to the ethics and local governance applications will occur and the appropriate training provided including Good Clinical Practice certification to ensure all researchers understand and abide by the protocol.

### 3.5. Follow-Up

Mortality will be prospectively identified through the routine notification of death following ED attendance in the last 30 days. Safety events will be prospectively identified when a safety event is logged and assigned to the ED as custodian for that event.

### 3.6. Data Management and Security

TECOR data will be recorded online using a REDCap data system licensed to the Department of Health, Tasmania. REDCap is a secure web application for building and managing online surveys and databases, free to consortium partners, secure, browser-based, metadata-driven designed by Vanderbilt University. Access to REDCap data entry will be restricted to onsite within the Tasmanian Health Service intranet and servers. Data will be preserved in this format for the duration of the project due to the prospective database design and may be used for future projects subject to independent and explicit ethical approval. This adheres to the strict privacy and confidentiality policies of the Tasmanian Health Service and relevant legislation and is supported by ethical approval. Due to these laws and regulations, the data gathered will not be available for open access and public re-use unless a separate ethical and governance application is made and approved. De-identified data will be extracted from REDCap for analysis in a variety of statistical programs including (but not limited to) Excel, STATA and R. Data will be stored for 15 years in accordance with Section 6 of the Disposal Schedule for Health Administration Records, DA2525 [53]. Following this time, they will be sent to the State Archives for disposal in accordance with the Archives Act 1983. Identified health information will not be transferred to another location for additional storage unless amendments are made and approved by the custodians of ethics and governance approvals.

### 3.7. Statistical Analysis

For reporting and publication purposes, all data will be exported in Microsoft Excel 2010 (Microsoft, Redmond, WA, USA) and analysed using SPSS PASW version 18.0 (SPSS, Inc., Chicago, IL, USA). Descriptive statistics will include the median and inter-quartile range (IQR from the 25th to the 75th percentile). T2 test or, as appropriate, exact tests will be used to compare groups of categorical data and to test for trends. For all analyses, actual *p*-values will be reported, and all tests will be two-tailed. Statistically significant differences will be considered at the $p < 0.05$ level, and 95% confidence intervals (CI) will be presented where possible.

## 4. Reporting

Data reporting will comply with local privacy laws. Data will be presented as aggregate data to ensure that information is not re-identifiable. Where reportable cases are 10 or less these will be aggregated or suppressed to maintain confidentiality.

Results will be included in a CQR feedback loop (Figure 2). Results will be disseminated to involved clinicians through departmental quality improvement and education programs. Staff in participating EDs will be informed of these reporting opportunities to assist in reinforcing the link between contributing clinical data and care outcomes. Further dissemination within the Tasmanian Health Service, Department of Health and the University of Tasmania will occur through scheduled education sessions such as hospital grand rounds and research symposia. Annual reporting is intended.

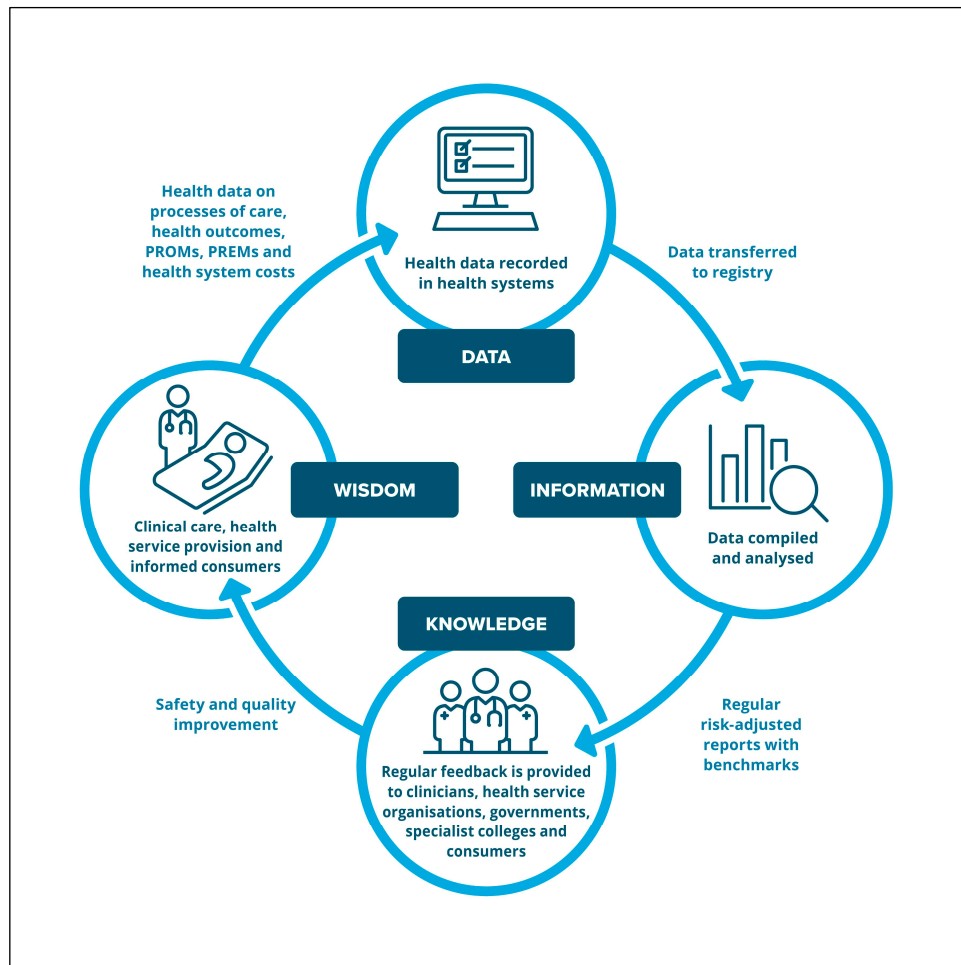

**Figure 2.** Australian Commission on Safety and Quality in Health Care clinical outcome feedback loop [54].

Results will also be used for research purposes where specific answers are able to be drawn from the registry data. These results will be published in appropriate scientific journals.

## 5. Discussion

Due to the largely private nature of funding of CQRs, the capacity for sophisticated registries is limited and they are therefore often based around disease or procedures where the patient population is not large (compared with the number of ED presentations for example) [55]. They may also be sponsored by diseases that have generous funding through not-for-profit organisations such as cancer, stroke and physiological trauma [24,34]. At present, there are few emergency care research registries in existence [36–38]. Due

to the inherently vulnerable environment of EDs, understanding the quality of care is paramount to improving the delivery of emergency care. Epidemiological studies in emergency care have been distracted by the intense focus on ED overcrowding, which is likely the primary cause of adverse outcomes in EDs. Other vulnerabilities in the health system should be explored to ensure that other causes of morbidity and mortality are not missed opportunities.

The study protocol described will address the important issues in identifying the degree of clinical safety and quality of emergency care delivered in Tasmania. In developing the TECOR, we have followed the guidance proposed in the ACSQHC framework for Australian clinical quality registries [20].

### 5.1. Future Studies: Nested Trials

Many studies either based on EDs or using ED data duplicate their data collection when performing disease-specific cohort studies such as survival in sudden cardiac arrest, accuracy of rapid antigen testing and prognostication of scoring systems [52–54]. The TECOR provides a framework for other observational studies through the collection of documented care to minimise this duplication. The data variables currently utilised in TECOR include a combination of routinely collected data for national reporting, baseline data considered essential to the primary objective as well as additional data points currently being collected through siloed observational studies (Table 2). It is the intention that the data collected through TECOR will contribute to other registries (with the requisite ethics and governance approvals) to minimise the current duplication and inefficiencies of collecting the same data variables multiple times over. For new research questions and quality improvement projects based in the EDs, and with additional ethics and governance amendments where required, TECOR will have the framework to add additional data variables in order to address specific research questions.

### 5.2. Limitations

The TECOR is specific in its data collection which can affect the comprehensiveness of its findings. Further to this, it may limit the questions asked of the registry for future studies. Data quality is also a limitation given the hybrid medical record keeping within the Tasmanian Health Service limiting large-scale data input into TECOR and increasing the prospect of human error. A significant limitation to all registries including TECOR is funding and sustainability. Finally, this registry is limited to Tasmania, which may not be reflective of a larger cohort.

## 6. Conclusions

The TECOR is expected to provide ongoing data on important processes of emergency care in Tasmania with the primary objective of determining the mortality and morbidity from emergency care. The registry initiative was designed to national standards and will meet the needs of the clinical community and impact the communities it serves.

**Author Contributions:** Conceptualization, V.T. and G.B.; methodology, V.T. and G.B.; formal analysis, V.T., G.B. and S.P.; investigation, V.T., G.B. and S.P.; resources, V.T., G.B. and S.P.; writing—original draft preparation, V.T.; writing—review and editing, V.T., G.B. and S.P.; project administration, S.P.; funding acquisition, V.T. All authors have read and agreed to the published version of the manuscript.

**Funding:** This research and the APC was funded by the Medical Research Future Fund (MRFF), grant number MRF2018041.

**Institutional Review Board Statement:** The study was conducted in accordance with the Declaration of Helsinki, and approved by the Ethics Committee of the University of Tasmania Human Research Ethics Committee (HREA30260, 26 February 2024).

**Informed Consent Statement:** Patient consent was waived and approved by the University of Tasmania Human Research Ethics Committee given that the information published will be de-identified and not re-identifiable and informed consent or opt-out consent for the number of patients expected is not feasible.

**Data Availability Statement:** The data presented in this study are available on request from the corresponding author due to local privacy laws.

**Conflicts of Interest:** The authors declare no conflicts of interest. The funders had no role in the design of the study; in the collection, analyses, or interpretation of data; in the writing of the manuscript; or in the decision to publish the results.

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
