# Peer review of "The Tasmanian Emergency Care Outcomes Registry (TECOR) Protocol"

_2813-7914, doi:10.3390/ecm1020017_

Round 1
Reviewer 1 Report
Comments and Suggestions for Authors
The study describes the establishment of the Tasmanian Emergency Care Outcomes Registry (TECOR) to monitor 19 emergency care processes and outcomes. The primary objective of TECOR is to monitor the unexpected 30-day mortality of patients who are cared for by the ED as well as 30-day safety events where emergency care was the primary contributor. Since this study is a protocol and did not provide data analysis, I have a few comments:
1. It would be useful to state whether these data will be open-access for public re-use.
2. Will these data be collected manually or electronically?
3. Some important points are missing in the dataset such as the hospital outcome, severity scores and so on. The granularity of the dataset is limited; there are some other examples of dataset establishment (PMID: 36690650 ), with high granularity for emergency care.
4. "Safety events will be prospectively identified when a safety event is logged and assigned to the ED as custodian for that event."---what specific events are safety events?
5. For Statistical Analysis section, there is lack of explicit objectives; what question do you want to solve?
Author Response
|
Comment |
Response |
|
1. It would be useful to state whether these data will be open-access for public re-use. |
Unfortunately, due to Tasmanian legislation, the data has not been given ethical or governance approval to be open access for public re-use. We have updated the manuscript under 3.6 with the following: “Due to these laws and regulations, the data gathered will not be available for open access and public re-use unless a separate ethical and governance application is made and approved.” |
|
2. Will these data be collected manually or electronically? |
Data collection will be electronic using the REDCap data system via semi-automated data input for routinely collected data and manual input into the electronic form where automation is not possible. We have added this line to 3.4 Data Collection: “Data will be collected via semi-automated importation into the registry for routinely collected data and manually inputted where automation is not possible.” |
|
3. Some important points are missing in the dataset such as the hospital outcome, severity scores and so on. The granularity of the dataset is limited; there are some other examples of dataset establishment (PMID: 36690650 ), with high granularity for emergency care. |
Thank you for the reference. As mentioned in 1.3 and 1.4, emergency care as opposed to critical care is in its infancy. Whilst internationally there are many critical care registries, such as the one that you have identified (PMID 3669-650), emergency care registries are lacking. We believe that the data points collected are appropriate for the aims of the registry, that is, to monitor the unexpected mortality of patients who are cared for by the emergency departments in Tasmania. |
|
4. "Safety events will be prospectively identified when a safety event is logged and assigned to the ED as custodian for that event."---what specific events are safety events? |
Thankyou for brining this to our attention. Under 1.5 we have described how safety events are identified in more detail by adding “The Tasmanian Health Services utilise a safety reporting and learning system as a non-punitive, confidential and voluntary program to collect and analyse safety concerns by healthcare staff.” |
|
5. For Statistical Analysis section, there is lack of explicit objectives; what question do you want to solve? |
As this is a registry protocol and as outlined in 2. Study Objectives; the main aim is to monitor the unexpected 30-day mortality of patients who are cared for by the ED as well as 30-day safety events where emergency care was the primary contributor. As such, data analysis will be largely descriptive. |

Reviewer 2 Report
Comments and Suggestions for Authors
This paper describes the protocol for the Tasmanian Emergency Outcomes Registry. It is generally well-written, describes the protocol quite clearly, and also highlights some of the common issues with quality care registers in Australia.
Abstract – well-written. Ln 21 – patients are cared for IN the ED, not by the ED (or by ED staff, whichever suits)
Introduction – Generally clearly written with good flow. Ln 44 – as not all patients are “resuscitated”, “management” may be a better word choice.
Lns 46-58 – for international readers, it would be helpful to clarify that not all EDs are staffed with each of the listed staff members. Smaller regional hospitals will not have immediate access to all of the medical staffing but it will be mix of those listed, particularly after hours.
Ln 80 – The references to unreliable coding using the ICD framework is very old – please use references that are more recent and Australian. Coding has improved since these papers were published almost 20 years ago and they are based on a very old classification version. It should also be clarified that the version of the ICD currently used in Australia (ICD-10-AM ) is a modification of the full ICD 10th revision.
Ln 120 – The Rural Acute Hospital Data Register (RAHDaR) includes emergency presentation data from regional and rural hospitals in South West Victoria and is another register in the peer-reviewed literature that may be worth assessing. Please check for other emergency data registers that may have been initially overlooked.
Ln 154 – “…submitted through a safety and learning reporting system software.”. Should this be “platform”? If not, the sentence needs rewording.
Methods – If a primary objective of TECOR is monitoring of 30-day safety events, why is the representation to ED timeframe 28 days? If there is a justification for the different timeframes, it needs to be included or the timeframes adjusted.
Ln 197 – “…as they are not by definition emergency departments”. This should include ACEM (with citation), whose definition I presume is being used.
Exclusion criteria – how are patients who have left with advice but no care provided dealt with? Also, if a person leaves before care is provided and then returns later that day/within 24 hrs, this appears to be excluded. This is also a very important issue to capture, particularly with ED overcrowding. These issues need to be clarified and/or justified.
Table 2 – The vast majority of data fields are included (mandatory) in the VEMD yet virtually none are noted as included in the table. The ICD-10-AM is now up to the eleventh edition and there have been a number of changes since the eight edition – is the eighth edition actually being used? If so, consideration should be given to updating to the eleventh edition to avoid data mismatches/quality issues or its use justified.
Reporting – Is there regular reporting/dissemination planned, or will it be on an ad hoc basis? Many registries include regular reporting schedules as an incentive to participate and also assist in maintaining data quality.
Discussion – Ln 300 Table 2 – this is the second table 2; should it actually be Table 1? It would be better placed in the introduction and should be referred to when appropriate in that section.
Ln 316 – “….TECOR will have the framework to add additional variables to prove the null hypothesis”. This should be changed as it is not an accurate statement – which null hypothesis? Will you be proving or investigating?
Conclusions – clear and concise
References – There are a number of errors in the formatting of the references. These must be addressed. Please remember to include access dates for digital maps, documents, etc.
Comments on the Quality of English LanguageThere are a few grammatical errors (most of which are noted above). Please do a final check.
Author Response
|
Comment |
Response |
|
Abstract – well-written. Ln 21 – patients are cared for IN the ED, not by the ED (or by ED staff, whichever suits) |
Thank you, this has been corrected to “in” |
|
Introduction – Generally clearly written with good flow. Ln 44 – as not all patients are “resuscitated”, “management” may be a better word choice. |
Thank you, resuscitated has been changed to managed |
|
Lns 46-58 – for international readers, it would be helpful to clarify that not all EDs are staffed with each of the listed staff members. Smaller regional hospitals will not have immediate access to all of the medical staffing but it will be mix of those listed, particularly after hours. |
Thank you for highlighting this – we have qualified our sentence by including “Medical staffing in larger regional and metropolitan EDs in Australia are typically 4 tiered...” |
|
Ln 80 – The references to unreliable coding using the ICD framework is very old – please use references that are more recent and Australian. Coding has improved since these papers were published almost 20 years ago and they are based on a very old classification version. It should also be clarified that the version of the ICD currently used in Australia (ICD-10-AM ) is a modification of the full ICD 10th revision. |
Thank you for highlighting the vintage of this assumption. We have removed the sentence implying that ICD-10-AM codes are inaccurate. For international readership, we have also included:
Any clinical data is limited to diagnosis. In Australia, this diagnosis classification is a derivation of the International Classification of Disease system, with Australian modifications currently up to its 12th revision. Further to this, in 2015 the Independent Health and Aged Care Pricing Authority (IHACPA) developed an abbreviated list for use in the ED known as the Emergency Department ICD-10-AM Principal Diagnosis Short List. |
|
Ln 120 – The Rural Acute Hospital Data Register (RAHDaR) includes emergency presentation data from regional and rural hospitals in South West Victoria and is another register in the peer-reviewed literature that may be worth assessing. Please check for other emergency data registers that may have been initially overlooked. |
Thank you for bringing this study to our attention. We have included this in our examples of emergency care registries, noting that they are examples and not a comprehensive list. |
|
Ln 154 – “…submitted through a safety and learning reporting system software.”. Should this be “platform”? If not, the sentence needs rewording. |
This has been modified as a result of another reviewers comments to:
The Tasmanian Health Services utilise a safety reporting and learning system as a non-punitive, confidential and voluntary program to collect and analyse safety concerns by healthcare staff. |
|
Methods – If a primary objective of TECOR is monitoring of 30-day safety events, why is the representation to ED timeframe 28 days? If there is a justification for the different timeframes, it needs to be included or the timeframes adjusted. |
Thank you, we have updated both the mortality and safety events to 28 days. |
|
Ln 197 – “…as they are not by definition emergency departments”. This should include ACEM (with citation), whose definition I presume is being used. |
The definition was described in the introduction and referenced to ACEM. We have also added the reference to this. |
|
Exclusion criteria – how are patients who have left with advice but no care provided dealt with? Also, if a person leaves before care is provided and then returns later that day/within 24 hrs, this appears to be excluded. This is also a very important issue to capture, particularly with ED overcrowding. These issues need to be clarified and/or justified. |
Our initial description of the exclusion criteria was not clear as the scenario relating to this exclusion were patients who were passing through the ED for patient identification labels before going to the ward, the cath lab etc. We have modified the sentence to better reflect this. It is our intention that all patients that did not wait would be included in our registry to understand these patterns and the risks associated. We have added a sentence to this effect as well:
Patients who administratively registered as presenting to an ED but where emergency care was not actually provided by ED Staff, such as inter hospital transfers that are required to enter the hospital via the ED for patient labels, will be excluded. Patients will also be excluded if they are not triaged as they will not have registered on the electronic system as being present. Patients who choose not to wait to be seen will be included if they are registered on the ED tracking system. |
|
Table 2 – The vast majority of data fields are included (mandatory) in the VEMD yet virtually none are noted as included in the table. The ICD-10-AM is now up to the eleventh edition and there have been a number of changes since the eight edition – is the eighth edition actually being used? If so, consideration should be given to updating to the eleventh edition to avoid data mismatches/quality issues or its use justified. |
Thank you for identifying this editing error. We have added a superscript 3 to all data points that are included in VEMD to confirm that virtually all are included in VEMD. We have updated the episode of care – “principle diagnosis, code” and “episode of care – secondary diagnoses and beyond, code” to include a superscript “5” that refers to “Tasmanian Health Service reported ICD-XX-AM, ICD-10-AM 12th edition at time of protocol publication” |
|
Reporting – Is there regular reporting/dissemination planned, or will it be on an ad hoc basis? Many registries include regular reporting schedules as an incentive to participate and also assist in maintaining data quality. |
It is our intention to report on an annual basis at this stage. This has also been added in the manuscript. |
|
Discussion – Ln 300 Table 2 – this is the second table 2; should it actually be Table 1? It would be better placed in the introduction and should be referred to when appropriate in that section. |
We have updated the first table 2 and renamed it table 1. We feel that this table 2. Is better suited in its current location as it is referenced in the discussion highlighting the significant duplication of data collection within EDs currently occurring with disease and procedure specific cohort studies/databases/registries. |
|
Ln 316 – “….TECOR will have the framework to add additional variables to prove the null hypothesis”. This should be changed as it is not an accurate statement – which null hypothesis? Will you be proving or investigating? |
We have updated the sentence to remove the null hypothesis – “data variables in order to address specific research questions.” |
|
Conclusions – clear and concise |
Thank you |
|
References – There are a number of errors in the formatting of the references. These must be addressed. Please remember to include access dates for digital maps, documents, etc. |
We have reviewed each reference and updated the formatting where we found errors. |

Reviewer 3 Report
Comments and Suggestions for Authors
Thank you for the opportunity to read this interesting manuscript. I ask the authors to take into account the following comments:
1) Do the authors consider supplementing the TECOR protocol with pre-hospital care? (did the patient come alone or was he brought by an ambulance? What pre-hospital assistance was provided?) - this may be important due to the patient's condition upon admission to the Emergency Department.
2) Do the authors consider including district hospitals in the program? Significant distances for patients requiring help in the middle of the country make it impossible to reach the Emergency Department. Maybe the program would indicate whether there is a need to create new Emergency Departments?
3) I ask the authors to complete the discussion and limitations of the TECOR program that they currently know or expect.
4) Only 21 items out of over 50 in the literature can be considered current (not older than 5 years). I recommend supplementing the bibliography with works relating to emergency care and emergency services, including:
a) Sosnowska-Mlak O, Curt N, Pinet-Peralta LM. Survival in sudden cardiac arrest in emergency room: case-control study. Crit. Care Innov. 2019; 2(3): 1-10. DOI: 10.32114/CCI.2019.2.3.1.10
b) Krishna PP, Velavarthipati RS, Srikanth M, Krishna BSG, Sriramula N, Goud DPK. To determine the prognostic accuracy of the HEART score as a predictor for major adverse cardiac events in patients presenting with chest pain to emergency department in a tertiary care hospital. Crit. Care Innov. 2023; 6(1): 1-16. DOI: 10.32114/CCI.2023.6.1.1.16
c) Leszczyński PK, Sobolewska P, Muraczyńska B, Gryz P, Kwapisz A. Impact of COVID-19 Pandemic on Quality of Health Services Provided by Emergency Medical Services and Emergency Departments in the Opinion of Patients: Pilot Study. Int J Environ Res Public Health. 2022; 19: 1232. DOI: 10.3390/ijerph19031232
Author Response
|
Comment |
Response |
|
1) Do the authors consider supplementing the TECOR protocol with pre-hospital care? (did the patient come alone or was he brought by an ambulance? What pre-hospital assistance was provided?) - this may be important due to the patient's condition upon admission to the Emergency Department. |
Thank you for asking this important question. The pre-hospital and ED interface is a crucial relationship. To this end, as per Table 2. We will include mode of arrival. It is beyond the scope of this registry, which is focused on ED performance and quality, to measure pre-hospital management. Furthermore, the pre-hospital system in Tasmania is performed by Ambulance Tasmania, which is a separate organisation and therefore their data is protected under different governance. |
|
2) Do the authors consider including district hospitals in the program? Significant distances for patients requiring help in the middle of the country make it impossible to reach the Emergency Department. Maybe the program would indicate whether there is a need to create new Emergency Departments? |
This is an important consideration that we hope the registry may be able to answer once it is up and running – the data points of “mode of arrival” and “postcode” will allow for subgroup analysis to understand geographical disadvantage as you mention. Unfortunately, it is not a high priority given that Tasmania is geographically quite small (68,401 km2 with population concentrations in only 4 areas as per Figure 1(a). |
|
3) I ask the authors to complete the discussion and limitations of the TECOR program that they currently know or expect. |
Thankyou for highlighting this area that was missing from the original manuscript. We have since updated to include:
5.2 Limitations The TECOR is specific in its data collection which can affect the comprehensiveness of its findings. Further to this, it may limit the questions asked of the registry for future studies. Data quality is also a limitation given the hybrid medical record keeping within the Tasmanian Health Service limiting large scale data input into TECOR and increasing the prospect of human error. A significant limitation to all registries including TECOR is funding and sustainability. Finally, this registry is limited to Tasmania, which may not be reflective of a larger cohort.
|
|
4) Only 21 items out of over 50 in the literature can be considered current (not older than 5 years). I recommend supplementing the bibliography with works relating to emergency care and emergency services, including:
a) Sosnowska-Mlak O, Curt N, Pinet-Peralta LM. Survival in sudden cardiac arrest in emergency room: case-control study. Crit. Care Innov. 2019; 2(3): 1-10. DOI: 10.32114/CCI.2019.2.3.1.10 b) Krishna PP, Velavarthipati RS, Srikanth M, Krishna BSG, Sriramula N, Goud DPK. To determine the prognostic accuracy of the HEART score as a predictor for major adverse cardiac events in patients presenting with chest pain to emergency department in a tertiary care hospital. Crit. Care Innov. 2023; 6(1): 1-16. DOI: 10.32114/CCI.2023.6.1.1.16
c) Leszczyński PK, Sobolewska P, Muraczyńska B, Gryz P, Kwapisz A. Impact of COVID-19 Pandemic on Quality of Health Services Provided by Emergency Medical Services and Emergency Departments in the Opinion of Patients: Pilot Study. Int J Environ Res Public Health. 2022; 19: 1232. DOI: 10.3390/ijerph19031232 |
Thankyou for highlighting the references. It is a reflection of the stagnant nature of emergency medicine research unfortunately within very little evidence around emergency care registries despite the need.
Thankyou for also highlighting articles you mentioned, although we note that these are all disease specific cohort studies rather than registries or databases. Nevertheless, they do point to the need for a registry to perhaps minimise the need for duplicitous data collection for individual cohort studies, similar to how the critical care registries internationally have evolved. We have therefore included these references in our manuscript in 5.1 |

Round 2
Reviewer 2 Report
Comments and Suggestions for Authors
Thank you for addressing the comments in the previous review. The paper is very well written and I hope to review futures papers generated from the project.
There are two small typos in the second version:
Ln 127 - the word "ref"
Ln 232 - "IDC-XX-AM"
